# Identifying metaphors towards physiotherapists

**Rabia Tugba Durdubas**[1]*, **Yildiz Yildirim**[2], **Hayri Baran Yosmaoglu**[3]

**1** Faculty of Physical Therapy and Rehabilitation, Hacettepe University, Ankara, Turkey, **2** Faculty of Education, Department of Measurement and Evaluation in Education, Aydin Adnan Menderes University, Aydin, Turkey, **3** Faculty of Health Sciences, Department of Physiotherapy and Rehabilitation, Baskent University, Ankara, Turkey

* rtugbakilic@gmail.com

**Data Availability Statement:** https://figshare.com/articles/dataset/_b_IDENTIFYING_METAPHORS_TOWARDS_PHYSIOTHERAPISTS_b_/26830105?file=48779716".

**Funding:** The author(s) received no specific funding for this work.

## Abstract

The aim of this study was to identify mental images that physiotherapists and student physiotherapists have regarding the concept of physiotherapist through metaphors. The study was a mixed-methods research that combined latent content analysis and categorical data analysis. Data were collected from 207 physiotherapists or physiotherapy students. Content analysis, a qualitative method, was performed on the raw data, which was composed of each participant's statement completing the following sentence "*Physiotherapist is like. . . because. . .*". Metaphors for physiotherapists were centered around 12 concepts. Most cited metaphor by student physiotherapists was "*healer*" (29.4%) and it was "*guiding*" (17.2%) for in-service physiotherapists. The rate of the "*negative connotation*" category in in-service physiotherapists (7.1%) was 2% higher than that of the student physiotherapists. Women mostly emphasized the "*healer*" (21.9%) category, while men mostly emphasized the "*healer*" and "*supporter*" (17.8%) categories. Participants aged under 25 and 26–35 age group were mostly in the "*healer*" category (26.4% and 18.8%, respectively) but the participants aged 36 years and over mostly opted for "*guiding*" and "*supporter*" (20.0%). There were fundamental differences between student physiotherapists and in-service physiotherapists in terms viewpoint regarding physiotherapy as a profession, and that these differences vary by gender and especially by age. During the undergraduate period, it will be beneficial to ensure that students have a realistic perspective with versatile training programs.

## Introduction

In the word metaphor, whose origin is based on the Greek word "metapherein", "meta" means "change" and "pherein" means "transfer" [1]. Metaphor was defined as a concept, fact, or event by analogy with another concept, fact, or event [2–5]. For this reason, metaphors appear in learning difficult-to-understand abstract concepts via analogies with known concepts [6]. Metaphors act as a bridge between newly learned knowledge and the existing knowledge [7].

The main functions of metaphors can be listed as follows: creating a mental model; emphasizing the similarities and differences; a blueprint and archetype of professional thought; and a powerful mental mapping and modeling mechanism for individuals to understand and

**Competing interests:** The authors have declared that no competing interests exist.

construct their own world; and, they are used as an important tool in education and training [8–16].

According to report published by the World Physiotherapy Confederation, the duties and responsibilities of physiotherapists can be summarized as follows; to make clinical decisions, developing a treatment plan, consulting with other healthcare professionals, implementing the treatment program, and, advising patients for self-regulation [17, 18].

Metaphor studies is very important in educational sciences. They have been conducted to reveal the ways of the perception of different concepts in different fields [15, 19–28]. There are publications on the perception of medicine and nursing [24, 29–31]. For example, nursing students use the metaphor of a "happy face" to describe nursing [32] while medical students use metaphors such as " water, constitution, life, human, mirror, and balance" to describe the concept of medical ethics [24]. As seen, metaphors in health sciences are important for their ability to convey professional perspectives in a simple and understandable manner, as well as for revealing the emotional and complex aspects of the profession that may go unnoticed in definitions [32–34]. However, to the best of our knowledge, regarding people's perceptions of the physiotherapist identity via metaphors, there exists no study in the literature.

From this point of view, it was aimed to identify the mental images that physiotherapists have about the concept of physiotherapist through metaphors, considering that it will contribute to the trainings, regulations, and studies to improve the profession of physiotherapy. At this point, the research question addressed by the study is;

- What metaphors do in-service and student physiotherapists use when describing their profession?

- Do these metaphors differ between in-service and student physiotherapists by experience, age, and gender?

## Methods

### Research model

In this study, it was aimed to identify in-service and student physiotherapists' perceptions of physiotherapist with the help of metaphors. This study is a mixed type research using qualitative and quantitative analysis methods. This method allowed that a better understanding of a specific phenomenon by using qualitative and quantitative methods together [35]. A mixed methods research combine both qualitative and quantitative analyzes while developing research that provides more depth than only qualitative or only quantitative research can produce [36].

*Research design*: A mixed method approach combining qualitative and quantitative methods was used [37]. In the qualitative part of the mixed-methods, a phenomenological research model was used. The phenomenological pattern deals with the phenomena that are recognized but cannot be understood in depth and detail. Phenomena can occur in various forms such as events, experiences, perceptions, orientations, concepts, and situations [38].

**Quantitative research design [39].** In the quantitative part of the mixed-methods, a descriptive research model was used. Because data converted from qualitative to quantitative were used to compare in-service and student physiotherapists, age and sex [40].

In this study, domain of COREQ criteria (research team and reflexivity, study design and analysis and findings) were examined and used in reporting the research [40].

The questionnaires for this study were administered from April 15, 2019, to November 20, 2019 in X (for blind review). Ethical approval of the study was obtained from X University (for

blind review) Ethics Committee (Date: 15.03.2019, #119). Informed consent was obtained from the participants through an online form. Participant approval was obtained through the question added to the beginning of the online submission form. The study did not include minor participants.

## Study group

The research study includes both in-service physiotherapists and physiotherapy students. The nature of the physiotherapy profession necessitates one-on-one care with patients or individuals in good health. Maintaining positive relationships with patients, caregivers, and other healthcare professionals is of utmost importance, especially in the demanding clinical setting and given the rigorous theoretical and practical coursework during the student years. Therefore, the perspectives of physiotherapy students about the profession, prior to gaining professional experience, have a greater potential for transformation following professional exposure compared to many other professions. We used purposive sampling method in the sample selection of the research. Purposive sampling allows in-depth research by selecting information-rich cases depending on the purpose of the study [41]. Because of the aim of this research is to identify and compare metaphors related students and in-service physiotherapists, we included these participants in the study according to the specified criterion [42, 43]. The criterion was to be in-service or student physiotherapists and to be volunteer. In this direction, the metaphors of 207 in-service and student physiotherapists were examined. But 57 metaphors were excluded from study group. Because some participants were neither student nor in-service physiotherapists or some of them were not complete the questionnaire. Finally, 150 metaphors were analyzed. The study group of the research consists of 51 students (%34) and 99 in service (%66). Of participants 105 were female (%70) and 45 were male (%30). Within the scope of the study, "I" abbreviation was used for in-service physiotherapists, and "S" abbreviation was used for student physiotherapists. In addition, each participant was given a number and represented by codes such as "S5" and "I2".

## Data collection tool

In the study, in order to identify the mental images about the concept of "*physiotherapist*", the participants were asked to complete this statement: "*Physiotherapist is like. . . because . . .*". The metaphor image specified with the concept of "*like*" in the first part of this expression pattern points out the link between the subject of the metaphor used and the source of the metaphor. In the "*because. . .*" part of the expression pattern, the participants provide a justification or a logical basis for this connection they have established in their own metaphors. An online questionnaire was prepared to identify the metaphors of the participants (metaphorical expression patterns) and delivered to in-service and student physiotherapists. These documents that were collected through voluntary participation constitute the main data source of the research. In addition, demographic information regarding the participants' ages, gender, active working status and class level for student physiotherapists was also obtained through the questionnaire. Because within the scope of the research, in-service and student physiotherapists, the variables of physiotherapists' age and gender were compared with each other.

## Data analysis

The data obtained in the research were analyzed by content analysis. [44] The latent content analysis was performed in this study. In qualitative content analysis, there is an effort to understand not only the manifest, but also the latent content of data [45]. The factist perspective is primarily the basis of qualitative content or thematic analyses, which are commonly used in

qualitative descriptive studies [46]. In this study, an inductive content analysis was conducted. Because the themes and categories were not determined before the analysis of the data, as in the thematic analysis, and categories and themes were created respectively from the codes. In inductive qualitative studies, any framework is data-derived for analysis [47].

The metaphors revealed by the participants and the conceptual categories created for these metaphors are presented in relation to the literature. To analyze the metaphors developed by the participants: a process consisting of five stages was followed: (1) coding and sorting, (2) compiling a sample metaphor image, (3) developing category, (4) establishing trustworthiness, [48] and (5) quantitative data analysis.

**Coding and sorting.** In the present study, 207 raw data were reached in total. The metaphor images created by the participants were arranged in an alphabetical order and a temporary list was created using these raw data. In the data collected, whether a certain metaphor was defined clearly was checked and the metaphors presented by each participant were coded (medicine, hope, mother, magician, etc.). At this stage, the answer sheets that did not specify any metaphor images, or those with the personal thoughts of the participants in general, instead of about metaphor images, were removed from the data set. Similarly, although some participants used a metaphor image, it was found that they left the "*because. . .*" part blank and did not present any logical basis for the metaphor in question. Based on these reasons, 57 metaphors that were determined to be missing of "because . . ." part were excluded from data analysis.

**Compilation of sample metaphor images.** A total of 150 valid metaphors were found in the present study, and 88 metaphor images in these metaphors were identified and listed. After the extraction process, the metaphor documents were revised and a "*sample metaphor statement*" considered to be representing each metaphor well was chosen. While it was seen that a common justification was expressed in some metaphors, in some other metaphor images, the justification after the "*because. . .*" part differed. For example, using the metaphor image of "mother" for the physiotherapist phenomenon, I14 uses the love aspect of the mother image as a logical basis with the expression "*unconditionally loves patients like mothers do*", and I12 highlights the "*healer*" image of mother by stating "*mothers try to heal their children*". Furthermore, more than one "*sample metaphor statement*" was created for some metaphors in which different logical reasons were proposed. Thus, a "*sample metaphor list*" was prepared for each of the 88 metaphor images along with the compilation of metaphor patterns that were supposed to represent them best. One of the reasons for the creation of this list was that there was a need for a source consisting of representative expressions to be used in the process of collecting metaphor images under a certain category. Another reason was to increase the validity of the data analysis and interpretation process of the research.

**Category development.** At this stage, the metaphor images produced by the participants were examined in terms of their common features regarding the physiotherapist phenomenon. In this process, how each metaphor image conceptualizes the phenomenon of physiotherapist was examined by considering the "*sample metaphor list*". To this end, each metaphor created by the participants was analyzed in terms of the subject of the metaphor, and the source of the metaphor [49]. Then, a total of 12 different conceptual categories were created by associating each metaphor image with a certain theme with its perspective of the physiotherapist. From this point of view, the metaphors used by the participants were grouped under 12 conceptual categories: "*guiding*", "*friendly*", "*patient/resilient*", "*who is needed*", "*surreal*", "*supporter/helping*", "*negative connotation*", "*healer*", "*versatile thinker*", "*someone connecting to life/giving hope*", "*someone adding meaning to life*", and "*someone working consistently and regularly*".

**Establishing trustworthiness.** Trustworthiness are two very important criteria in terms of ensuring the credibility of the research results. One of the important criteria to ensure

validity in a qualitative study is to report in detail the data and to explain how the results of the study were achieved [44]. To this end, the steps followed in the data analysis process were explained in detail in order to ensure the internal validity of the results of the research. In addition, for each of the 88 different metaphor images obtained in the study, sample metaphor patterns that were considered to represent them best were compiled and these metaphor patterns were included in the findings section. In order to ensure the reliability of the research, the opinion of an expert who had previously conducted research on metaphor and education was consulted. This expert was asked to classify the metaphors in most appropriate 12 conceptual categories. Then, the classifications made by the expert and the researcher were compared. At this stage, between these two coder's agreements was used Miles and Huberman's (1994) formula (Agreement percentage = Agreement / Agreement + Disagreement) for the trustworthiness evidence. In order to ensure trustworthiness in qualitative studies, the agreement between expert and researcher evaluations for the categories is expected to be at least 80% [49, 50]. As a result of the calculations made in this research, it was found that there was a 90% agreement. To eliminate the remaining 10% disagreement, two coders reviewed the data together and ensured agreement between them.

The backgrounds of the authors have influenced the study design and hypothesis. One of the authors, with over 20 years of experience as a professor at a physiotherapy school, conducted research on student success in education and exam types [51]. Additionally, they participated in a multi-center, comprehensive project examining unemployment concerns [52] and business awareness [53]. Furthermore, in collaboration with the other author of this study, they were involved in research examining the field competence of physiotherapists [54]. The research question regarding whether the students' perspective on the profession, shaped during these studies, influenced their motivation for education, laid the foundation for this study in the future.

**Quantitative data analysis.** In this study, after the development of 12 conceptual categories, all data were transferred to the SPSS 24 statistical program. Then, the number (n) and percentage (%) of the participants representing 150 metaphors and 12 conceptual categories were calculated. In addition, the Chi-Square (2) test was conducted in variables such as gender, age, and the status of being an in-service physiotherapist or student physiotherapist. Qualitative and quantitative data types can be converted into each other for the purpose of examining the target phenomenon. Because qualitative and quantitative data are not different from each other [40]. Therefore, data convertetion was carried out by quantifying the qualitative data collected in this mixed-methods research.

## Results

The metaphor images produced by the in-service and student physiotherapists about the concept of "*physiotherapist*" and the findings regarding the conceptual categories were presented below.

### Findings on metaphors of in-service and student physiotherapists about their profession

There was a total of 150 valid metaphors related to the concept of physiotherapist. In these metaphors, 88 different metaphor images were used, and these metaphor images were collected under 12 different conceptual categories as follows (Table 1). Also, the word cloud that was drawn for the metaphor images were presented in Fig 1.

**Physiotherapist as a guide.** This category involves 20 (13.3%) in-service and student physiotherapists and 14 metaphor images. The guide (f = 3), pathfinder (f = 4), and mother

Table 1. Distribution of metaphors by categories.

| Categories | n (%) | Metaphor Image | f |
|---|---|---|---|
| 1. Guiding | 20 (13.3) | Guide (3), Pathfinder (4), Sun (1), Light source (1), Marathon-runner (1), Teacher (1), Sculptor (1), Mother (2), Light (1), Life coach (1), Street lamp (1), Advisor (1), Star (1), Operator (1). | 14 |
| 2. Friend | 9 (6.0) | Psychologist (1), Friend (1), Mother (2), Joker (1), Family member (1), Parent (1), Friend (1), Sister (1). | 8 |
| 3. Patient/resilient | 3 (2.0) | Patience (1), Rock (1), Diamond (1) | 3 |
| 4. Someone needed | 8 (5.3) | Smartphone (1), Mortar (1), Horizon line (1), Hero (1), Our feet and hands (1), Healer (1), Cane (1), Water (1). | 8 |
| 5. Surreal | 5 (3.3) | Magic (1), Magician (1), Hopeful dreams (1), Superman (1), Artist (1). | 5 |
| 6. Supporter/helping | 19 (12.6) | Right-hand man (1), Life coach (2), Angel (2), Crutches (1), Plane-tree (1), Mother (3), Support (1), Step (1), The most important part of the team (1), Family (1), Hand reaching out to the patient (1), Sidekick (1), Assistant (1), Psychologist (1), Empathy (1). | 15 |
| 7. Negative connotation | 8 (5.3) | Relative (1), Distant relative (1), Assistant health personnel (1), Scapegoat (1), Porter (1), Machine (1), Donkey (2). | 7 |
| 8. Healer | 31 (20.6) | Medicine (4), Healing (1), Master of function (1), Hope (1), Sea (1), Sculptor (2), Repairman (2), Repairer (1), Water (1), Building foundation (1), Superhero (1), Traffic police (1), Healing (4), Mother (3), Physiotherapist (1), Magic hand (1), Doctor (3), Light (1), Pain reliever (1). | 19 |
| 9. Versatile thinker | 9 (6.0) | Guide (1), Sculptor (1), Mother (2), Swiss army knife (2), Architect (1), Physician (1), Life coach (1). | 7 |
| 10. Someone connecting to life/ giving hope | 18 (12.0) | Wing (1), Psychologist (1), Life (1), Tailor (1), Hope (5), Hero (1), Life saver (2), Herald of spring (1), Sun (2), Tree (1), Life support (1), Life water (1). | 12 |
| 11. Someone adding meaning to life | 9 (6.0) | Life (3), Life coach (1), Doctor's helping hand (1), Cream (1), Water (1), Magician (1), Mother's hand (1). | 7 |
| 12. Someone working consistently and regularly | 11 (7.3) | Mother (2), Artist (1), Diamond (1), Iron (1), Comrade (2), Savior (1), Clock (1), Bee (1), Architect (1). | 9 |

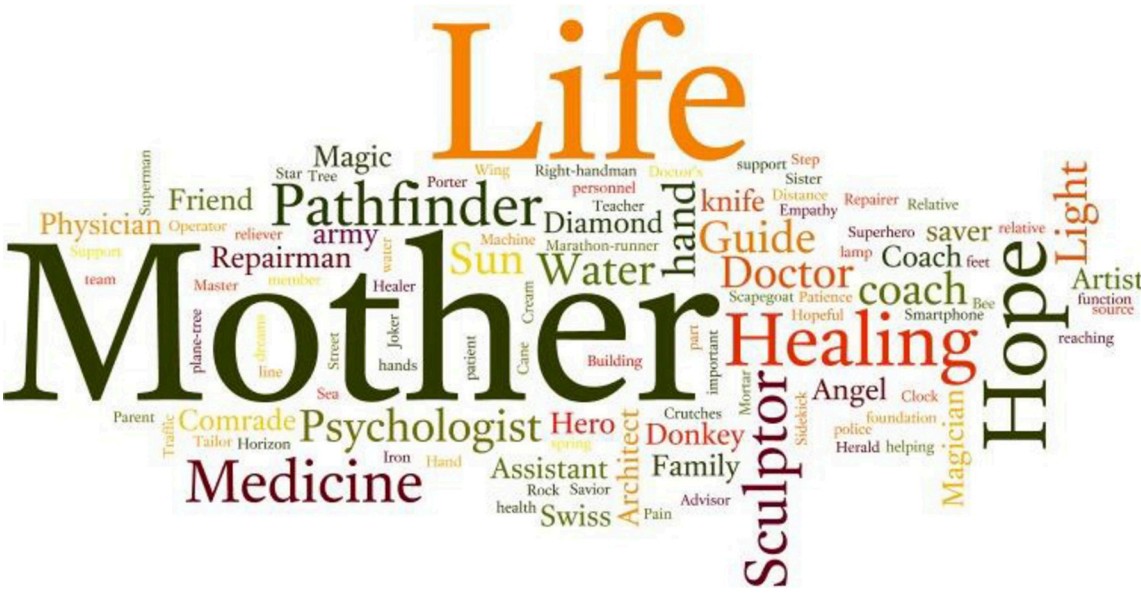

**Fig 1. The word cloud for metaphor images.**

(f = 2) metaphor images occurred more frequently, and other metaphor images were mentioned once. One of the participants, I163, stated for the metaphor image of the "*guide*" that "*the physiotherapist educates and guides patients regarding which way to go regarding their complaints*". For the "*guiding*" metaphor image in this category, I156 said: "*In many cases, the physiotherapist shows the way and methods of getting health to the person who consults him/her.*" Regarding the metaphor image of the mother, the following statement was said: "*the physiotherapist teaches everything again*" (S118).

**Physiotherapist as a friend.**   Regarding the physiotherapist as a friend category, the followings were considered: friendship, closeness, and kinship relations. The sincerity and reliability of aspects of the concept of friendship were highlighted. In this context, the *"wild card/ joker''* (a person who is good at everything) metaphor image was explained as follows: ". . . *The physiotherapist is like a son, a sibling, or sometimes a confidant*" (I156). The metaphor image of "*family member*" was expressed as follows: "*Most of the time, the physiotherapist looks after the patient more than the family; he is a brother or sister who constantly thinks about how to improve the patient even at night*" (S151).

**Physiotherapist as someone patient/resilient.**   In this category, there are three metaphor images: '*patient as stone/very patient*' (f = 1)', '*diamond (f = 1)*' and '*rock (f = 1)*'. One of the participants, I161, explained his/her perspective towards the physiotherapist with the metaphor image of 'rock' as follows: "*the physiotherapist's work area is tiring, his colleagues are challenging, working conditions are hard. Therefore, she/he must be like a rock in order to survive.*" For the "*diamond*" metaphor image, a participant stated: "*at least as durable as diamond*" (I19).

**Physiotherapist as someone needed.**   The physiotherapist as someone needed category includes 8 (5.3%) participants and 8 different metaphor images. While a participant expressed the physiotherapist in a "*healer*" metaphor image as "*Everyone needs a physiotherapist*" (S181), another participant explained the "*water*" metaphor image as "You need it at every moment of your life. " (I184)

**Physiotherapist as a surreal being.**   In the physiotherapist as a surreal being category, for the "*magician*" metaphor image, the following justification was given by S108: "*The moment he touches, he returns everything to normal. It gives happiness not only to the sick but to everyone.*" For *"dreams full of hope"*, which is another metaphor image in this category, this was said: "*Because the physiotherapist does his/her best for you to perform the functions that you thought you could never do*" (S85).

**Physiotherapist as supporting/helping.**   In this category, "*life coach*" (f = 2) metaphor image was the leading one. The participants explained the "*life coach*" metaphor image as "*the physiotherapist should be able to support the patient from every angle*" (S64) and "*because the physiotherapist helps the patient and the individual in all kinds of matters*" (S117). For the metaphor image of "*supporter*" in this category, the following expression was used: "*a supporter who can understand best both the patient and the patient's caregivers*" (S87).

**Physiotherapist as negative connotation.**   In the negative connotation conceptual category, the metaphor image of "*donkey*" (f = 2) was repeated. The following expressions were used for this metaphor image: "*because the physiotherapist works and works, does not get a proper salary, and does not get the value he/she deserves*" (I139) and "*no matter how hard the physiotherapist works, he/she cannot gain much*" (I140). The metaphor image of the "*machine*" expressed as negative opinion about the physiotherapist as follows: "*because it is the most oppressed occupational group among the gears of the sector*" (I106).

**Physiotherapist as healer.**   The conceptual category of physiotherapist as healer constitutes 20.6% (f = 31) of the participants. The most frequently recurring metaphor images in this category are 'medicine' (f = 4) and "*healing*" (f = 4). These expressions were used for the metaphor image of "*medicine*": "*the physiotherapist approaches the injury like a medicine given by the*

*doctor and heals the injury*" (I182), "*the physiotherapist is a remedy for troubles*" (S207), "*the physiotherapist heals the patients* (S147) ", and "*the physiotherapist heals*" (I189). The metaphor image of "*healing*" in this category was explained as: "*the physiotherapist is a means of healing using only his/her hands*" (I89); and "*the physiotherapist helps improve or maintain our health*" (S37). In addition to these, the other metaphor images that were frequently repeated were "*doctor*" (f = 3) and "*mother*" (f = 3). The following expressions were used for the metaphor image of "*doctor*": "*the physiotherapist is the person authorized to distribute healing*" (I100) and "*the physiotherapist heals the patient*" (S59). These expressions were used for the "*mother*" metaphor image: "*mothers try to heal their children*" (I199) and "*the physiotherapist always wants the good of people and tries to heal them*" (S38).

**Physiotherapist as versatile thinker.** The metaphor image that stands out in the versatile thinker conceptual category is the analogy of "*Swedish army knife*" (f = 2). This expression was used for the "*Swedish army knife*" metaphor image: "*the physiotherapist can approach different problems differently; s/he has a wide range of solutions and seeks to solve problems in multiple ways; s/he has a broad perspective*" (S71). The metaphor image of the "*mother*" was explained as: "*the physiotherapist takes responsibility in order to reach the physical, mental, and social well-being of the patients*" (I13).

**Physiotherapist as one who connects to life and gives hope.** In the conceptual category that one connects to life and gives hope, the metaphor image of "*hope*" (f = 5) was the most frequently repeated metaphor. The following expressions were used regarding the metaphor image of "*hope*": "*while thinking that everything is over, everything starts again thanks to the physiotherapist*" and "*it is an orthopedic patient's hopes that the physiotherapist ends his/her pain*" (S116). S97 made the following explanation for the "*life-saving*" metaphor image in this category: "*the physiotherapist is a blessing given to people who despair of life.*" The metaphor image of the "*sun*" was explained as follows: "*the physiotherapist, s/he is illumination of the darkness in the patient's world with his/her light, and s/he will be a ray of hope for the patient. S/he warms and enlightens the world of those who are cold with illness*" (I132).

**Physiotherapist who adds meaning to life.** In the conceptual category of one adds meaning to life, the metaphor image of "*life*" (f = 3) was repeated frequently. I157 expressed the metaphor of "*life"* as follows: "*because the physiotherapist adds meaning to life.*", and I95 expressed as follows: "*the physiotherapist makes life livable.*" The following expression was used for the metaphor image of "*mother's hand*": "*physiotherapist provides a beautiful and meaningful to life with his/her treatment*" (I197). The "*cream*" metaphor image was explained as follows: "*without the physiotherapist, the patient can continue his/her life, but there is no comfort in life*" (I122).

**Physiotherapist as someone working consistently and regularly.** The metaphor images of "*mother*" (f = 2) and "*comrade*" (f = 2) were repeated in the conceptual category of someone working consistently and regularly. The following expression was used for the metaphor image of "*mother*": "*the physiotherapist treats his/her patient best for a long time*" (S11). For the "*comrade*" metaphor image, the following expression was used: "*The physiotherapist walks a long way with the patients in order to bring them to where they want to be*" (S130). The "*clock*" image in this category was explained as follows: "*because the physiotherapist treats on an hourly basis. They work regularly*" (S70).

## Findings on metaphors regarding sex, age, and the status of being an in-service or student physiotherapist

Chi-square analysis results regarding the relationships between conceptual categories and participants' sex, age, and the status of being an in-service or student physiotherapist were presented in Table 2.

**Table 2. Chi-square test results between conceptual categories sex, age, and the status of being an in-service or student physiotherapist.**

| Category | | Student/In-Service Physiotherapist* | | Sex* | | Age* | | |
|---|---|---|---|---|---|---|---|---|
| | | Student | In-Service | Female | Male | ≤25 | 26–35 | ≥36 |
| 1. Guiding | n | 3 | 17 | 17 | 3 | 7 | 7 | 6 |
| | (%) | (5.9) | (17.2) | (16.2) | (6.7) | (9.7) | (14.6) | (20.0) |
| 2. Friend | n | 2 | 7 | 6 | 3 | 7 | 2 | 0 |
| | (%) | (3.9) | (7.1) | (5.7) | (6.7) | (9.7) | (4.2) | (0.0) |
| 3. Patient/Resilient | n | 0 | 3 | 2 | 1 | 0 | 3 | 0 |
| | (%) | (0.0) | (3.0) | (1.9) | (2.2) | (0.0) | (6.3) | (0.0) |
| 4. Someone needed | n | 3 | 5 | 6 | 2 | 2 | 3 | 3 |
| | (%) | (5.9) | (5.1) | (5.7) | (4.4) | (2.8) | (6.3) | (10.0) |
| 5. Surreal | n | 3 | 2 | 5 | 0 | 4 | 0 | 1 |
| | (%) | (5.9) | (2.0) | (4.8) | (0.0) | (5.6) | (0.0) | (3.3) |
| 6. Supporter/Helping | n | 7 | 12 | 11 | 8 | 8 | 5 | 6 |
| | (%) | (13.7) | (12.1) | (10.5) | (17.8) | (11.1) | (10.4) | (20.0 |
| 7. Negative Connotation | n | 1 | 7 | 4 | 4 | 3 | 3 | 2 |
| | (%) | (2.0) | (7.1) | (3.8) | (8.9) | (4.2) | (6.3) | (6.7) |
| 8. Healer | n | 15 | 16 | 23 | 8 | 19 | 9 | 3 |
| | (%) | (29.4) | (16.2) | (21.9) | (17.8) | (26.4) | (18.8) | (10.0) |
| 9. Versatile thinker | n | 3 | 6 | 4 | 6 | 4 | 3 | 2 |
| | (%) | (5.9) | (6.1) | (3.8) | (11.1) | (5.6) | (6.3) | (6.7) |
| 10. Someone connecting to life/giving hope | n | 9 | 9 | 13 | 5 | 12 | 4 | 2 |
| | (%) | (17.6) | (9.1) | (12.4) | (11.1) | (16.7) | (8.3) | (6.7) |
| 11. Someone adding meaning to life | n | 2 | 7 | 6 | 3 | 3 | 4 | 2 |
| | (%) | (3.9) | (7.1) | (5.7) | (6.7) | (4.2) | (8.3) | (6.7) |
| 12. Someone working consistently and regularly | n | 3 | 8 | 8 | 3 | 3 | 5 | 3 |
| | (%) | (5.9) | (8.1) | (7.6) | (6.7) | (4.2) | (10.4) | (10.0) |
| Total | n | 51 | 99 | 105 | 45 | 72 | 48 | 30 |
| | (%) | (100) | (100) | (100) | (100) | (100) | (100) | (100) |

* The number of cells whose expected frequency falls below 5 constitutes approximately 41.7% of the total cells according to the variable of "in-service/student" (n = 10) and "gender" (n = 10); and constitutes approximately 72.2% of the total cells according to the variable of "age" (n = 26). For this reason meaningfulness interpretation was not made in this analysis, and only cross table was used.

Table 2 shows that the student physiotherapists proposed metaphors mostly in the "*healer*" (29.4%) category, which was followed by the metaphors in the categories of "*someone connecting to life/giving hope*" (17.6%) and "*supporter-helping*" (13.7%), in this order. In addition, student physiotherapists proposed very few metaphors in the "*negative connotation*" (2.0%) category. An analysis of the distribution of metaphors proposed by in-service physiotherapists by the conceptual categories showed that, unlike student physiotherapists, more metaphors were created in the conceptual category of "*guiding*" (17.2%), which was followed by "*healer*" (16.2%) and "*supporter/helping*" (12.1%), in this order; and, in-service physiotherapists proposed more "*negative connotation*" metaphors (7.1%) than student physiotherapists did (2.0%).

An analysis of the distribution of the conceptual categories by gender showed that women mostly proposed metaphors in the category of "*healer*" (21.9%), which was followed by "*guiding*" (16.2%) and "*someone connecting to life/giving hope*" (12.4%). The female participants placed the least emphasis on the "*patient/resilient*" (1.9%) aspect of the physiotherapist. An

analysis of the metaphors of the male participants showed that the categories of "*healer*" (17.8%) and "*supporter/helping*" (17.8%) were equally emphasized. Furthermore, it was observed that the categories that demonstrate the characteristics of the physiotherapist as "*someone connecting to life/giving hope*" (11.1%) and "*versatile thinker*" (11.1%) were both heavily emphasized. Although the "*guiding*" category was prominent in the metaphors of the female participants, it was found that the rate of male participants proposing in this category was quite low (6.7%). Moreover, unlike women, the male participants did not propose any metaphors in the "*surreal*" category.

The distribution of conceptual categories by age variable was also examined in the study. Participants aged 25 years and under mostly proposed a metaphor in the "*healer*" (26.4%) category. Moreover, the metaphors in the categories of "*connecting to life/giving hope*" (16.7%) and "*supporter/helping*" (11.1%) were also mostly in this age group. Similarly, the category of "*healer*" (18.8%) had a great number of the metaphors form the 26–35 age group. However, it was observed that the rate of metaphors emphasizing the "*guiding*" feature of the physiotherapist (14.6%) was higher in the 26–35 age group than the participants aged 25 years and less. Participants aged 36 years and over proposed metaphors mostly in the categories of "*guiding*" (20.0%) and "*supporter/helping*" (20.0%). Although it was not possible to make interpretation regarding the statistical significance of the difference, it was observed that as the age increased, the rate of the participants' metaphors in the "*guiding*" category also increased, but the rate in the "healer" and "*someone connecting to life/giving hope*" categories decreased.

## Discussion

In the present study, the mental images that in-service and student physiotherapists have regarding the concept of physiotherapist through metaphors was determined. Then, a latent content analysis was used on the raw data. Consequently, metaphors reflecting the participants' perspectives regarding the physiotherapist were identified in relation with their age, sex, and status of being an in-service or student physiotherapist.

The primary finding of this study was that student physiotherapists predominantly created metaphors within the "healer" category, whereas practicing physiotherapists tended to generate metaphors in the "guiding" conceptual category. This outcome is likely due to the fact that, during their education, physiotherapy students are taught that treatments administered by physiotherapists aim to improve a single factor. However, as they gain experience in the field, they come to understand that physiotherapy treatments are multifactorial, encompassing biopsychosocial and holistic approaches, which play a crucial role in treatment [53]. Over time and with experience, physiotherapists realize that treatment should be multifaceted, addressing both physical and emotional aspects [53]. Consequently, they view their role as pivotal in guiding and educating patients through recovery steps, rather than just focusing on the recovery itself, leading them to construct metaphors primarily within the "guiding" conceptual category.

Another important finding of the study is that the mental images acquired regarding the concept of physiotherapist are affected by gender. While the women proposed metaphors mostly in the categories of "*healer*", "*guiding*", and "*connecting to life*", in this order; men proposed metaphors mostly in "*healer*" and "*supporter/helper*" categories equally, which were followed by the categories of "*connecting to life/giving hope*" and "*versatile*". The reason for this difference may be the effect of gender difference and reflection of the roles of the individuals in society. The concepts of sex and gender concepts can be defined biologically and socially. Basically, sex refers to the biological differences of men and women. On the other hand, the concept of being a woman or a man in daily life, which is formed by the society according to the biological sex of the person, refers to gender. In other words, while sex defines the

anatomical difference between men and women; gender refers to the social construction difference between men and women [55]. There are many studies on gender roles and patterns. Research shows that men are expected to be strong and able to support their families, and women, on the other hand, are expected to be patient, understanding, being able to look after home/family, and organizing human relations. It has been observed that there are the expectations for male and female roles in the traditional society and family system [56, 57]. Research findings on the roles of women and men from different countries present similar results. While men were attributed to features such as strong, rude, adventurous, aggressive, dominant, independent, and self-confident; and women are attributed features such as dependent, weak, polite, emotional, sensitive, talkative, caring, and helpful [58]. The gender roles and stereotypes that society expects the individual to adopt affect the way they perceive themselves [59]. We think that the difference in the imageries between men and women in the present study and the prominence of the "*supporter/helper*" metaphor as well as the "*guiding*" metaphor in male participants may be the result of male individuals being affected by gender roles.

The study findings indicated that the distribution of conceptual categories varied with age. Participants aged 25 and under, as well as those aged 26–35, primarily focused on the "healer" category. However, the proportion of 26-35-year-old participants who used the "guiding" metaphor was higher than that of the younger group. Participants aged 36 and over mainly proposed metaphors in the "guiding" and "supporter/helping" categories. These findings suggest that with increasing age, the use of "guiding" metaphors also increased, while the use of "healer" and "connecting to life/giving hope" metaphors decreased. Aging is both a social and biological category [59]. It has been studied from various perspectives by disciplines such as medicine, psychology, sociology, and anthropology. Understanding the meaning of aging is crucial, and philosophy contributes to this understanding. From Plato's viewpoint, old age is a period of wisdom [60]. Studies [61, 62] that view aging as a maturation period and suggest that people become more knowledgeable and virtuous explain the higher frequency of "guiding" metaphors among older participants in this study. To the best of our knowledge, there exists no metaphor study about the concept of physiotherapist in the literature. However, in a metaphor study related to the concept of "*physician*", it was observed that all the metaphor categories contained only positive features and positive perceptions. The physician candidates mostly associated their physician identity with the "sacrifice" category, which was followed by "*wisdom*", "*compulsory/required*", "*creativity*", and "*leadership*" categories, in this order [63]. These categories are similar to the ones found in the present study: "*guiding*", "*friendly*", "*patient/resilient*", "*who is needed*", "*supporter/helping*", "*healer*", "*versatile thinker*", "*someone connecting to life/giving hope*", "*someone adding meaning to life*", and "*someone working consistently and regularly*". Unlike the study about "*physicians*", the present study contained "*negative connotation*" metaphor images for the physiotherapist. It was determined that the students formed a very little metaphor for the concept of physiotherapist in the "*negative association*" category; however, it was found that this ratio was higher in in-service physiotherapists compared to student physiotherapists. To this end, working conditions after graduation, job status, the reputation of being a physiotherapist, professional regulations, and other disciplines' point of view regarding the field of physiotherapy are thought to contribute to the formation of the differences in this imagery.

The strength of this study lies in its incorporation of both qualitative and quantitative methodological aspects. Consequently, the study not only uncovers the identified metaphors associated with physiotherapists but also examines how age, gender, and business experience impact the alteration of these metaphors. However, it's important to note that the study has limitations. It does not consider into other personal, sociocultural, and economic factors that might influence physiotherapists' perspectives on their profession.

## Conclusion

There were fundamental differences between student physiotherapists and in-service physiotherapists in terms viewpoint regarding physiotherapy as a profession, and that these differences vary by gender and especially by age. For instance, the use of "healer" and "machine" metaphors by physiotherapy students in their professional descriptions, while experienced physiotherapists prefer "supporter" and "donkey" metaphors in response, suggests that these two groups perceive the challenges of their profession differently. In addition to professional education courses, adding curriculum related to challenges in the workplace, such as communication, time management, and community psychology, could be beneficial for physiotherapy students. Furthermore, during the undergraduate period, it will be beneficial to ensure that students have a realistic perspective with versatile training programs improving working conditions via occupational laws and regulations may also be beneficial in order to increase high working satisfaction rates and more optimist point of view can be obtained.

## Author Contributions

**Conceptualization:** Rabia Tugba Durdubas.

**Data curation:** Rabia Tugba Durdubas.

**Formal analysis:** Rabia Tugba Durdubas.

**Methodology:** Rabia Tugba Durdubas.

**Resources:** Rabia Tugba Durdubas.

**Software:** Rabia Tugba Durdubas.

**Supervision:** Rabia Tugba Durdubas.

**Validation:** Rabia Tugba Durdubas.

**Writing – original draft:** Rabia Tugba Durdubas.

**Writing – review & editing:** Yildiz Yildirim, Hayri Baran Yosmaoglu.

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
