## [Decision Letter · Decision Letter 0]

5 Sep 2023

PONE-D-23-19512A QUALITATIVE STUDY ON IDENTIFYING METAPHORS TOWARDS PHYSIOTHERAPISTSPLOS ONE

Dear Dr. Kilic,

Thank you for submitting your manuscript to PLOS ONE. After careful consideration, we feel that it has merit but does not fully meet PLOS ONE’s publication criteria as it currently stands. Therefore, we invite you to submit a revised version of the manuscript that addresses the points raised during the review process. Please submit your revised manuscript by 18th October 2023. If you will need more time than this to complete your revisions, please reply to this message or contact the journal office at plosone@plos.org. Please include the following items when submitting your revised manuscript:A rebuttal letter that responds to each point raised by the academic editor and reviewer(s). You should upload this letter as a separate file labeled 'Response to Reviewers'.A marked-up copy of your manuscript that highlights changes made to the original version. You should upload this as a separate file labeled 'Revised Manuscript with Track Changes'.An unmarked version of your revised paper without tracked changes. You should upload this as a separate file labeled 'Manuscript'.

We look forward to receiving your revised manuscript.

Kind regards,

Jibril Mohammed, BSc, MSc, PhD

Academic Editor

PLOS ONE

Journal Requirements:

Reviewers' comments:

Reviewer's Responses to Questions

**Comments to the Author**

1. Is the manuscript technically sound, and do the data support the conclusions?

Reviewer #1: Partly

Reviewer #2: Yes

2. Has the statistical analysis been performed appropriately and rigorously? 

Reviewer #1: N/A

Reviewer #2: N/A

3. Have the authors made all data underlying the findings in their manuscript fully available?

Reviewer #1: No

Reviewer #2: Yes

4. Is the manuscript presented in an intelligible fashion and written in standard English?

Reviewer #1: Yes

Reviewer #2: Yes

5. Review Comments to the Author

Reviewer #1: Thank you for the opportunity to review your interesting manuscript. My major concern is the designed employed, phenomenology. What was described in the methods does not reflect any of the three phenomenological approaches, descriptive, interpretative, or hermeneutic. I suggest you provide a nuanced explanantion and a clear justification of the designed you employed and the choices you made. From what I can deduce, the work presented here is more in congruence with any of qualitative content analysis/generic qualitative approach/qualitative descriptive design. Many critical aspects of the method were not presented such as study context, philosophy behind the research design, and reflexivity statement of the authors. Kindly use the consolidated criteria for reporting qualitative research (COREQ) as a guide. I have also made some suggestion for you to consider and provided feedback throughout the manuscript. Kindly see the attachment. Thank you.

Reviewer #2: Comment

Abstract

I wonder how 207 participants were used. Was there no point when theoretical saturation was achieved with some of the participants during the data collection?

In addition, it is not conventional to have the percentage of respondents to the questions. That’s quantitative! In qualitative, we are concerned about the views and experiences as they narrated by the study population.

Introduction

Please can clearly state the relevance of metaphors in physiotherapy practice? Please give examples also- be it in medicine or nursing; or even a hypothetical example in physiotherapy. The concept ‘metaphor’ rather seems abstract in your introduction.

Method

Please how did you determine that particular metaphors are week and discarded them?

Result

Please refer to my comments on the abstract.

Discussion

First and the second paragraphs of your discussion did not have any citations. And it is not conventional for a whole paragraph not to carry a single citation. Please structure the paragraphs in such a way they will have at least one citation each.

Conclusion

Please look at the conclusion again. It is only comparing fundamental differences between student physiotherapists and in-service physiotherapists in terms viewpoint regarding physiotherapy as a profession, which is not the primary aim of your study and which contravenes the principles of a qualitative research. Please revisit this. Second paragraph of the discussion section captures what can be used as your conclusion.

6. PLOS authors have the option to publish the peer review history of their article (what does this mean?). If published, this will include your full peer review and any attached files.

Reviewer #1: **Yes: **Dr. Surajo Kamilu Sulaiman

Reviewer #2: **Yes: **Auwal Abdullahi

---

## [Author Response · Author response to Decision Letter 0]

2 Jan 2024

RESPONSE LETTER

1. Please provide additional details regarding participant consent. In the ethics statement in the Methods and online submission information, please ensure that you have specified what type you obtained (for instance, written or verbal, and if verbal, how it was documented and witnessed). 

The following sentence was added to the Research model section to clarify the issue: 

‘Participant approval was obtained through the question added to the beginning of the online submission form.’

2. If your study included minors, state whether you obtained consent from parents or guardians. If the need for consent was waived by the ethics committee, please include this information.

The following sentence was added to the Research model section to clarify the issue: 

‘The study did not include minor participants.’

We did not include such information because we did not perform a retrospective study of medical records or archived samples.

---

## [Decision Letter · Decision Letter 1]

3 Jun 2024

PONE-D-23-19512R1IDENTIFYING METAPHORS TOWARDS PHYSIOTHERAPISTSPLOS ONE

Dear Dr. Kilic,

Thank you for submitting your manuscript to PLOS ONE. After careful consideration, we feel that it has merit but does not fully meet PLOS ONE’s publication criteria as it currently stands. Therefore, we invite you to submit a revised version of the manuscript that addresses the points raised during the review process.

We look forward to receiving your revised manuscript.

Kind regards,

Michal Ptaszynski, PhD

Academic Editor

PLOS ONE

Journal Requirements:

Reviewers' comments:

Reviewer's Responses to Questions

**Comments to the Author**

1. If the authors have adequately addressed your comments raised in a previous round of review and you feel that this manuscript is now acceptable for publication, you may indicate that here to bypass the “Comments to the Author” section, enter your conflict of interest statement in the “Confidential to Editor” section, and submit your "Accept" recommendation.

Reviewer #1: All comments have been addressed

Reviewer #3: (No Response)

2. Is the manuscript technically sound, and do the data support the conclusions?

Reviewer #1: Yes

Reviewer #3: Yes

3. Has the statistical analysis been performed appropriately and rigorously? 

Reviewer #1: Yes

Reviewer #3: Yes

4. Have the authors made all data underlying the findings in their manuscript fully available?

Reviewer #1: Yes

Reviewer #3: Yes

5. Is the manuscript presented in an intelligible fashion and written in standard English?

Reviewer #1: Yes

Reviewer #3: Yes

6. Review Comments to the Author

Reviewer #1: Thank you for the opportunity to read the revised manuscript. My previous comments and observations were addressed. However, there is a minor issue in the present draft. The sentence in line 407-408 seems incomplete, kindly modify.

Reviewer #3: Dear all

I realize that authors have many journals to consider when they want to publish their work, so I appreciate your interest in PLOS ONE; I am very happy to be able to write in a positive way. It is evident that you have put a great deal of effort into this project and I want to praise your efforts. Fortunately, the actual contribution from your study is clear and, the manuscript as currently written suggests that it might be suitable for sharing information about this topic, but the paper that you reported, needs a few minor edits. I should like to thank you for giving me an opportunity to consider this work for publication.

It may be that the you would like to consider resubmitting it, in which case I hope that the comments from my review may help you to revise it before resubmitting it. These comments are given below.

Best Regards

General review:

- The topic described in the paper is very interesting;

Introduction section:

- Some references are missing in many sentences;

Methods section:

- No calculation or reference was given for how the sample size was calculated;

- No reporting statement was described or clearly reported that supported the type of study;

Discussion section:

- Discussions should be reviewed in light of the overall improvement of the paper. Redundant sentences and prewritten information should be avoided. Focus on take-home messages.

7. PLOS authors have the option to publish the peer review history of their article (what does this mean?). If published, this will include your full peer review and any attached files.

Reviewer #1: **Yes: **Dr. Surajo Kamilu Sulaiman

Reviewer #3: No

---

## [Author Response · Author response to Decision Letter 1]

26 Aug 2024

Reviewer #1: 

1. Thank you for the opportunity to read the revised manuscript. My previous comments and observations were addressed. However, there is a minor issue in the present draft. The sentence in line 407-408 seems incomplete, kindly modify.

Answer: The sentence on lines 407 and 408 has been changed as follows.

‘In the present study, the mental images that in-service and student physiotherapists have regarding the concept of physiotherapist through metaphors was determined’.

Reviewer #3: 

1. Introduction section:

- Some references are missing in many sentences;

Answer: References were added to the sentences where references were missing.

2. Methods section:

a. No calculation or reference was given for how the sample size was calculated;

The sampling method has already been described in the study group section of the article as follows: 

We used purposive sampling method in the sample selection of the research. Purposive sampling allows in-depth research by selecting information-rich cases depending on the purpose of the study [43]. Because of the aim of this research is to identify and compare metaphors related students and in-service physiotherapists, we included these participants in the study according to the specified criterion [44, 45]. The criterion was to be in-service or student physiotherapists and to be volunteer. 

b. No reporting statement was described or clearly reported that supported the type of study;

The mixed method of the research model is stated in the research model section of the manuscript. Additionally, it was emphasized that the qualitative part of the research was phenomenology and the quantitative part was a descriptive research. Type of the study was described in main text as follows:

In this study, it was aimed to identify in-service and student physiotherapists’ perceptions of physiotherapist with the help of metaphors. This study is a mixed type research using qualitative and quantitative analysis methods.. This method allowed that a better understanding of a specific phenomenon by using qualitative and quantitative methods together [34]. A mixed methods research combine both qualitative and quantitative analyzes while developing research that provides more depth than only qualitative or only quantitative research can produce [35].

3. Discussion section:

- Discussions should be reviewed in light of the overall improvement of the paper. Redundant sentences and prewritten information should be avoided. Focus on take-home messages.

Answer: The discussion has been reviewed in light of the overall development of the article. and necessary arrangements were made.

---

## [Decision Letter · Decision Letter 2]

17 Sep 2024

IDENTIFYING METAPHORS TOWARDS PHYSIOTHERAPISTS

PONE-D-23-19512R2

Dear Dr. Kilic,

We’re pleased to inform you that your manuscript has been judged scientifically suitable for publication and will be formally accepted for publication once it meets all outstanding technical requirements.

Kind regards,

Michal Ptaszynski, PhD

Academic Editor

PLOS ONE

Additional Editor Comments (optional):

Reviewers' comments:

Reviewer's Responses to Questions

**Comments to the Author**

1. If the authors have adequately addressed your comments raised in a previous round of review and you feel that this manuscript is now acceptable for publication, you may indicate that here to bypass the “Comments to the Author” section, enter your conflict of interest statement in the “Confidential to Editor” section, and submit your "Accept" recommendation.

Reviewer #3: All comments have been addressed

2. Is the manuscript technically sound, and do the data support the conclusions?

Reviewer #3: Yes

3. Has the statistical analysis been performed appropriately and rigorously? 

Reviewer #3: Yes

4. Have the authors made all data underlying the findings in their manuscript fully available?

Reviewer #3: Yes

5. Is the manuscript presented in an intelligible fashion and written in standard English?

Reviewer #3: Yes

6. Review Comments to the Author

Reviewer #3: Dear Authors

I would like to thank you for giving me an opportunity to consider this work for publication. You well done the a point by point answer to the comments of the reviewers.

7. PLOS authors have the option to publish the peer review history of their article (what does this mean?). If published, this will include your full peer review and any attached files.

Reviewer #3: No

---

## [Editor Report · Acceptance letter]

14 Oct 2024

PONE-D-23-19512R2 

PLOS ONE

Dear Dr. DURDUBAS, 

I'm pleased to inform you that your manuscript has been deemed suitable for publication in PLOS ONE. Congratulations! Your manuscript is now being handed over to our production team.

Kind regards, 

on behalf of

Dr. Michal Ptaszynski 

Academic Editor

PLOS ONE